# LGR-Net: A Lightweight Defect Detection Network Aimed at Elevator Guide Rail Pressure Plates

**DOI:** 10.3390/s25061702

**Published:** 2025-03-10

**Authors:** Ruizhen Gao, Meng Chen, Yue Pan, Jiaxin Zhang, Haipeng Zhang, Ziyue Zhao

**Affiliations:** 1School of Mechanical Engineering and Equipment, Hebei University of Engineering, Handan 056038, China; gaoruizhen@hebeu.edu.cn (R.G.); 18796309572@163.com (M.C.); panyue8728@163.com (Y.P.); 18132301036@163.com (J.Z.); 15512777173@163.com (H.Z.); 2Key Laboratory of Intelligent Industrial Equipment Technology of Hebei Province, Hebei University of Engineering, Handan 056038, China; 3Collaborative Innovation Center for Modern Equipment Manufacturing of Jinan New Area (Hebei), Hebei University of Engineering, Handan 056038, China; 4School of Information and Electrical Engineering, Hebei University of Engineering, Handan 056038, China

**Keywords:** guide rail pressure plate, small defects, MobileNetV3, GhostConv, CBAM

## Abstract

In elevator systems, pressure plates secure guide rails and limit displacement, but defects compromise their performance under stress. Current detection algorithms face challenges in achieving high localization accuracy and computational efficiency when detecting small defects in guide rail pressure plates. To overcome these limitations, this paper proposes a lightweight defect detection network (LGR-Net) for guide rail pressure plates based on the YOLOv8n algorithm. To solve the problem of excessive model parameters in the original algorithm, we enhance the baseline model’s backbone network by incorporating the lightweight MobileNetV3 and optimize the neck network using the Ghost convolution module (GhostConv). To improve the localization accuracy for small defects, we add a high-resolution small object detection layer (P2 layer) and integrate the Convolutional Block Attention Module (CBAM) to construct a four-scale feature fusion network. This study employs various data augmentation methods to construct a custom dataset for guide rail pressure plate defect detection. The experimental results show that LGR-Net outperforms other YOLO-series models in terms of overall performance, achieving optimal results in terms of precision (*p* = 98.7%), recall (R = 98.9%), mAP (99.4%), and parameter count (2,412,118). LGR-Net achieves low computational complexity and high detection accuracy, providing an efficient and effective solution for defect detection in elevator guide rail pressure plates.

## 1. Introduction

Guide rail clamps are pivotal components within elevator systems, tasked with securing and constraining the movement of guide rails (illustrated in Figure 1a,b). Their integrity is essential for safeguarding the elevator’s operational safety [1]. Nonetheless, the manufacturing process of these clamps is frequently marred by pitting corrosion, which begets a multitude of surface cavities (depicted in Figure 1c,d). Such imperfections markedly compromise the clamps’ mechanical robustness. In service, elevators are subjected to variable stresses stemming from load imbalances, shaft subsidence, and thermal fluctuations. Surface flaws on the clamps can undermine their capacity to withstand these forces, thereby elevating the potential for mishaps [2]. Hence, the imperative of defect scrutiny in the fabrication of guide rail clamps cannot be overstated. The evolution of image processing technologies and the advent of superior imaging apparatus have propelled the widespread adoption of machine vision for the scrutiny and categorization of workpiece anomalies [3]. In this vein, deep learning-infused machine vision strategies for defect detection have surfaced [4]. The deployment of deep neural network architectures for the autonomous extraction of image characteristics [5] facilitates the proficient discernment and classification of defects in guide rail clamps. This advancement markedly bolsters detection precision and efficacy, curtails manual expenditure, and augments the comprehensive capability of the detection framework.

Thanks to the rapid development and significant performance improvement of GPUs, training large deep neural networks has become increasingly practical [6]. Convolutional neural network (CNN)-based object detection algorithms have rapidly developed and gradually replaced traditional detection methods in image analysis [7]. CNN-based object detection algorithms are typically divided into two categories. One category follows traditional object detection methods, represented by the R-CNN series [8,9,10], which uses a two-stage network. In a two-stage network, region proposals are first generated, followed by CNN-based feature extraction, classification, and bounding box regression. Therefore, the prolonged time required to process different components has become a key limiting factor for its real-time applications. The other category includes single-stage networks, represented by YOLO [11,12,13,14] and SSD [15], which are based on global regression or classification. These networks directly map image pixels to bounding box coordinates and class probabilities, thereby reducing computational overhead. YOLO is known for its fast detection speed and strong generalization ability, offering a balance between accuracy and speed, which is why it is widely used in object detection [16]. However, the YOLO algorithm still faces challenges in defect detection [17].

Firstly, in contrast to the overall structure of the clamp, minor surface defects on guide rail clamps occupy a relatively small pixel area in images and contain limited semantic information. The YOLO algorithm’s feature extraction network performs significant downsampling on the image, and the successive convolution and pooling operations progressively erode these feature details [18]. This is the primary reason why the YOLO algorithm struggles to accurately identify minor defects on the surface of guide rail clamps. Secondly, the YOLO-series algorithms still involve a substantial number of parameters, making real-time detection challenging for edge devices without GPU support [19]. To address these limitations, this paper proposes a lightweight defect detection network for guide rail clamps, termed LGR-Net. The main contributions are as follows:(1)A P2 detection layer is introduced to optimize the combination of feature scales. By employing a four-scale feature fusion approach, low-level detail information and high-level semantic information in guide rail clamp images are effectively integrated, significantly enhancing the network’s capability to detect minor defects.(2)The CBAM is incorporated during the four-scale feature fusion stage to strengthen the extraction of channel and spatial feature information, further improving the network’s accuracy in identifying minor defects.(3)MobileNetV3 is adopted as the feature extraction backbone, and the Ghost module is introduced to optimize the convolutional structure of the feature fusion network, resulting in the construction of a C2fGhost module. This design ensures high detection accuracy while substantially reducing computational costs.(4)This paper constructs a self-built dataset for elevator guide rail clamp defects and proposes a lightweight detection network, LGR-Net, based on the YOLOv8 algorithm. The proposed network not only significantly reduces the number of parameters in the detection algorithm but also enhances its accuracy in detecting minor defects on guide rail clamps. Furthermore, it enables the deployment of the algorithm on edge devices.

## 2. Related Work

### 2.1. Defect Detection Method Based on Deep Learning

To tackle the problem of the standard PAN (Path Aggregation Network) losing local details and positional information in deeper layers, Zhang et al. [20] proposed a lightweight and detail-sensitive PAN. This method incorporates a feature transformation module embedded with an attention mechanism and optimizes lightweight implementation, specifically targeting small object detection in multi-scale defect scenarios. Shao et al. [21] developed a semantic information interaction module, which combines deep and shallow semantic features and introduces an interaction mechanism between fused features and shallow features to improve the detection of minor defects. Cui et al. [22] utilized a multi-level feature fusion approach to enhance the performance of steel surface defect detection. By integrating a lightweight attention enhancement module, this method effectively fuses shallow and deep feature maps, thereby boosting the expressive capability of the feature maps. Hao et al. [23] adopted a crosslayer fusion strategy for semantic and spatial features, providing richer information across different scales and enabling more accurate predictions for small object detection tasks. Yi et al. [24] introduced a deep information feature fusion module, which seamlessly integrates high-level and low-level features within the YOLOv7 feature extraction network, significantly improving the model’s precision in detecting small target defects. Liu [25] combined multi-head self-attention mechanisms and adaptive attention mechanisms with local spatial attention modules to construct a global attention module, optimizing the feature extraction network of YOLOv8 and enhancing the model’s capability to handle unstructured defects.

### 2.2. Lightweight Deep Neural Network

The goal of neural network lightweighting is to reduce the computational load, storage requirements, and power consumption while maintaining model performance, thereby enhancing operational efficiency and deployment flexibility on resource-constrained devices [26]. Researchers have made significant progress in exploring lightweighting techniques for DCNN (Deep Convolutional Neural Network) models. Zhong et al. [25] proposed a dual-convolution kernel scheme to construct a lightweight deep neural network. This scheme employs 3 × 3 and 1 × 1 convolutions in parallel and reduces the number of input channels processed by each kernel through a group convolution strategy, significantly decreasing the number of parameters compared to standard convolution. Shen et al. [27] introduced the R-ECA module, which replaces traditional convolution with depthwise-separable convolution and facilitates cross-channel information interaction using 1 × 1 convolution, thereby substantially reducing the computational load and parameter count while preserving the richness of feature learning. Chen et al. [28] utilized neural architecture search (NAS) to automatically design lightweight networks adapted to multi-scale features and transferred prior knowledge from larger models to lightweight ones for detecting defects in photovoltaic cells. Lei et al. [29] developed DGNet, a lightweight defect detection model for battery current collectors. DGNet combines an adaptive lightweight backbone network (DOS module) with a lightweight feature fusion network (GS_FPN), leveraging DOConv and Shufflenet V2 for feature extraction while reducing parameter redundancy and improving detection accuracy. Qu et al. [30] optimized the feature extraction network of YOLOv8 using multi-scale group convolution (MSGConv). By dividing input channels into four groups and applying different-sized convolution kernels to each group, they significantly reduced parameter computations.

### 2.3. Overview of YOLOv8 Model

The YOLOv8 algorithm is an advanced object detection model developed by the Ultralytics team, representing the latest advancements in the YOLO series of algorithms. This algorithm redefines the object detection task as a single regression problem, enabling the network to efficiently predict multiple bounding boxes and class probabilities simultaneously, thereby enhancing both detection speed and accuracy. YOLOv8 introduces an anchor-free detection method that decouples the classification and regression tasks through its head structure, thereby reducing interference arising from the number and selection of anchor boxes. Furthermore, YOLOv8 employs a lightweight C2f module that replaces the traditional C3 module, providing richer gradient flow information during feature extraction and contributing to improved detection accuracy. In terms of positive and negative sample allocation, YOLOv8 utilizes an adaptive Task-Aligned Assigner strategy, which dynamically adjusts the sample allocation ratio based on the training performance to optimize the detection results. Regarding the loss function, YOLOv8 combines various loss calculation methods for classification and regression tasks, ensuring rapid convergence and high flexibility. The network structure of the algorithm (as shown in Figure 2) consists of four components, the input layer, backbone network, feature fusion network, and prediction network, making it suitable for various applications, including object detection, instance segmentation, and image classification. This architecture demonstrates robust scalability and practicality, establishing YOLOv8 as a new benchmark in the field of computer vision.

## 3. Methods

To address the limitations of traditional networks in accurately identifying small defects on guide rail pressure plates, along with their high computational cost, this paper proposes the LGR-Net detection algorithm (as shown in Figure 3). The LGR-Net network is built upon the YOLOv8n baseline model and incorporates several improvements. The backbone network, being a critical component for feature extraction, initially has a high parameter count. To address this issue, this study designs a lightweight backbone network based on MobileNetV3, reducing the parameter count by approximately 20% compared to the original network. Furthermore, the Ghost convolution module and a newly designed C2fGhost module are introduced to optimize the neck network, reducing the parameter count by an additional 4%. Finally, to enhance the algorithm’s ability to detect small defects, a small object detection layer is added, and a four-scale feature fusion network is constructed. Additionally, the CBAM is incorporated to improve the model’s focus on small defect detection.

### 3.1. Optimization of Output-Layer Feature Scale Combinations

YOLOv8 employs three feature detection layers to process feature maps at different scales, catering to the detection needs of objects of varying sizes. The P3 detection layer extracts features from shallower convolutional layers, preserving a higher resolution to capture the fine contours and intricate details of small-sized objects. The P5 detection layer, situated in the deeper layers of the YOLOv8 network, derives more abstract and comprehensive semantic features through multiple convolution and pooling operations, demonstrating superior performance in detecting large-sized objects. The P4 detection layer, positioned between P3 and P5, strikes a balance between detail information and semantic features, making it particularly effective for detecting medium-sized objects. While the P3 detection layer is capable of identifying small objects, its performance in detecting tiny objects remains limited. This limitation stems from the phenomenon of feature disappearance in the detection layer after continuous convolution processing of objects with extremely low pixel proportions. Specifically, in images of size 640 × 640, small objects typically occupy areas ranging from 8 × 8 to 32 × 32 pixels, whereas tiny objects with areas smaller than 8 × 8 pixels lose their feature information in the P3 layer during downsampling. To address this issue, this paper introduces an additional, shallower P2 feature detection layer, constructing a four-scale feature fusion network based on the pixel dimensions of a series of tiny hole defects. This approach effectively mitigates the loss of detailed features during network downsampling caused by the minimal proportion of tiny defects in the image.

### 3.2. CBAM 

Although the P2 detection layer can capture more detailed features, it may also amplify background noise, leading the model to misclassify noise as defects. Particularly in low-contrast or complex-textured backgrounds, the features of tiny defects may still be insufficiently prominent, making it difficult for the model to accurately identify these defects. To address this, this paper incorporates the Convolutional Block Attention Module (CBAM) [31] after the P2 detection layer. By leveraging its dual mechanisms of a channel attention module (CAM) and a spatial attention module (SAM) (as shown in Figure 4), CBAM dynamically adjusts the weights of important channels and spatial regions in the feature map, thereby enhancing the model’s ability to detect tiny defects.

The primary role of the channel attention module (CAM) is to assign weights to each channel within the feature map, amplifying those linked to minor defects while diminishing the influence of unrelated channels. As depicted in Figure 5, the CAM operates through a series of structured steps. Initially, in the feature extraction stage, it utilizes both average pooling and max pooling techniques to gather comprehensive information across each channel. Average pooling is instrumental in mapping the general spread of defects, whereas max pooling accentuates the distinctive local characteristics of these defects. Subsequently, during the channel weight computation phase, the outcomes from both pooling methods are channeled into a shared multilayer perceptron (MLP). This MLP intelligently recalibrates the channel weights in response to the nuanced edge variations of the defects, thereby facilitating a dynamic refinement of the feature map. In the concluding feature map weighting phase, the computed channel weights are integrated with the original feature map through multiplication, thereby elevating the visibility of minor defects within the map. This adaptive mechanism empowers the CAM to direct the model’s attention more effectively towards the pivotal features indicative of defects.

The spatial attention module (SAM) dynamically modulates the weight assigned to each spatial position within the feature map, thereby enabling the model to accurately pinpoint defects. Illustrated in Figure 6, the SAM’s architecture begins by consolidating spatial information through average pooling and max pooling operations applied across the channel dimension, resulting in the creation of two distinct 2D feature maps. These maps are then concatenated and fed into a convolutional layer, which synthesizes a spatial weight map. This map is subsequently normalized to a [0, 1] range via a sigmoid function. In the final step, the SAM integrates this spatial weight map with the original feature map through multiplication, effectively enabling the network to diminish the significance of non-critical areas while accentuating regions of importance. This mechanism is particularly effective when analyzing guide rail clamp imagery, as the SAM allocates greater emphasis to minuscule hole defects—spanning just a few pixels—by assigning them higher weights within the spatial weight map. This not only amplifies the characteristics of defect-laden regions but also facilitates the model’s ability to precisely identify the locations of these defects.

### 3.3. Improvements to the Backbone Module

The MobileNet [32,33,34] series of lightweight convolutional neural networks is specifically designed for mobile and embedded devices. Its third-generation model, MobileNetV3 (as shown in Figure 7), incorporates neural architecture search (NAS) technology, depthwise-separable convolution, the Squeeze-and-Excitation (SE) attention mechanism, and the h-swish activation function. These enhancements strike an optimal balance between computational efficiency and accuracy, making MobileNetV3 an excellent candidate for optimizing the YOLOv8 backbone network.

Firstly, MobileNetV3 utilizes depthwise-separable convolution, which decomposes standard convolution into depthwise convolution and pointwise convolution. This design significantly reduces the computational complexity and parameter count, providing a notable advantage in lightweight applications. In each depthwise convolution layer, spatial convolutions are applied exclusively to individual channels, while pointwise convolution employs a 1 × 1 kernel to integrate cross-channel information. Compared to standard convolution, this approach preserves the model’s expressive power while dramatically lowering computational overhead.

Secondly, the network employs an inverted residual structure that efficiently represents features by alternately expanding and compressing them along the channel dimension. Skip connections are utilized to retain critical information. By leveraging NAS (neural architecture search) technology, the network selects the optimal architecture from numerous candidate designs, achieving superior performance across diverse tasks.

Thirdly, MobileNetV3 integrates the SE (Squeeze-and-Excitation) attention mechanism into its inverted residual blocks. The SE module captures channel-level contextual information via global pooling and amplifies the focus on key features through weight learning, further enhancing the network’s perception and representational ability.

Lastly, MobileNetV3 introduces the h-swish activation function, replacing the Sigmoid function with a piecewise linear ReLU6 function. This substitution eliminates exponential calculations, significantly reducing the computational complexity.(1)$$h-swish\left(x\right)=x\frac{ReLU6\left(x+3\right)}{6}$$(2)$$ReLU6\left(x\right)=min(\max\left(0,x\right),6)$$

### 3.4. Neck Module Improvements

GhostNet [35] is a lightweight network designed by Huawei’s Noah’s Ark Lab in 2020, with GhostConv being a convolution module within GhostNet that can replace traditional convolutions. The core working principle of the Ghost module lies in reducing redundant computations in convolution operations to generate “Ghost Feature Maps” (as shown in Figure 8). This approach significantly reduces computational costs while maintaining model performance. Specifically, traditional convolution operations typically rely on a large number of filters to generate feature maps. In contrast, the Ghost module divides this process into two steps: First, a small number of filters are used to generate a limited set of “Intrinsic Feature Maps”. Then, the Ghost module applies inexpensive linear transformation operations (e.g., depthwise convolution) to these intrinsic feature maps to generate “Ghost Feature Maps”, compensating for any missing information and achieving a representation similar to that of the original convolution layer. The core computational formula of the Ghost module is expressed as follows:(3)Y=X+Φ(X)

Among these, *X* represents the intrinsic feature map, while Φ(*X*) denotes the ghost feature maps generated by inexpensive linear operations (depthwise convolution, translation, affine transformation, etc.). The final output feature map, *Y*, consists of two components: the intrinsic feature map and the ghost feature maps.

The C2fGhost module is a novel design that improves upon the original C2f module in the network. It achieves significant reductions in computational complexity by replacing all bottleneck structures in the C2f module with GhostBottlenecks. A GhostBottleneck first expands the number of channels using the GhostConv module, which is subsequently processed through normalization and activation functions. Finally, another GhostConv operation reduces the number of channels to match the input dimensions. The feature maps are further optimized through residual path fusion, enhancing feature learning capabilities. Additionally, C2fGhost employs cross-stage feature fusion and gradient truncation techniques to increase the diversity of learned features across network layers, effectively reducing redundant gradient information and improving the network’s learning performance. By minimizing the number of 3 × 3 convolution operations, this module compresses the model size and parameter count, making it particularly suitable for real-time processing.

### 3.5. Small Object Detection Layer

Due to the small size of small-object samples and the relatively large downsampling factor of YOLOv8, deeper feature maps struggle to capture the feature information of small objects. Consequently, the original YOLOv8 model exhibits poor detection performance for small objects. The input image size of the original model is 640 × 640, and the maximum detection scale is 80 × 80, where each grid has a receptive field of 8 × 8. If the height and width of the target in the original image are both smaller than 8 pixels, the original network struggles to identify the target’s features within the grid.

To address this limitation, this paper introduces a 160 × 160 small object detection layer into the original network, including additional fusion feature layers and an extra detection head to enhance the semantic information and feature representation of small objects. Specifically, the 80 × 80 feature layer from the fifth layer of the backbone is stacked with the upsampled feature layer from the neck. This stacked feature map undergoes C2f and upsampling operations, resulting in a deep semantic feature layer containing small-object features. It is then further stacked with the shallow positional feature layer from the third layer of the backbone, creating a comprehensive 160 × 160 fusion feature layer that improves the representation of both semantic features and positional information for small objects. Finally, the fused feature layer is processed through C2fGhost and delivered to an additional decoupled detection head in the head module.

## 4. Experiments and Result Analysis

### 4.1. Dataset and Data Processing

Since no publicly accessible datasets for guide rail clamps are available, this study relied on a custom-built dataset for experimentation, consisting of 1000 images captured from multiple angles and positions of guide rail clamps. As depicted in Figure 9, the dataset encompasses a variety of defects, including pitting holes, cracks, scratches, and wear. The data annotation process was carried out using the open-source labeling tool LabelImg. Five distinct labels were established for annotation: W represents the guide rail clamp body, d denotes pitting hole defects, c indicates crack defects, s signifies scratch defects, and w represents wear defects.

To further bolster the model’s generalization capabilities, data augmentation techniques were employed during training. These techniques included random rotation, scaling, flipping, brightness adjustment, and noise addition, as illustrated in Figure 10. Such augmentation strategies simulate a wide range of real-world conditions, enhancing the model’s ability to detect defects across different angles, sizes, and environmental variations. This, in turn, improves the robustness and accuracy of defect detection. Following data augmentation, the dataset expanded to 2000 images, which were then randomly partitioned into training, validation, and test sets in an 8:1:1 ratio. This resulted in 1600 images for training and 200 images each for validation and testing.

### 4.2. Experimental Environment and Parameter Settings

The experimental operating system employed in this study was Ubuntu 20.04, with PyTorch 2.4.1 serving as the foundational framework for the developed deep learning model. The detailed experimental environment is presented in Table 1. The hyperparameters used during the model training phase are presented in Table 2. The results of the training are illustrated in Figure 11.

### 4.3. Evaluation Metrics

To comprehensively assess the performance of the model, this study introduces four key metrics: precision (P), recall (R), Average Precision (AP), and Mean Average Precision (mAP). Precision (P) refers to the proportion of actual positive samples among all samples predicted as positive by the model, serving as a measure of the model’s accuracy. In contrast, recall (R) indicates the proportion of correctly predicted positive samples among all actual positive samples.

The metrics are defined as follows:(4)$$P=\frac{TP}{TP+FP}$$(5)$$R=\frac{TP}{TP+FN}$$(6)$$AP=\int_{0}^{1}P\;\left(R\right)\;dR$$(7)$$mAP=\frac{1}{n}\sum_{i=1}^{n}{AP}_{i}$$
where
TP denotes True Positives;FN represents False Negatives;FP indicates False Positives;AP is the Average Precision for a single class label;n is the total number of classes.

### 4.4. Ablation Experiments

To validate the performance improvements brought about by incorporating the MobileNetV3 module, GhostConv module, small object detection layer, and CBAM into the original YOLOv8 model, this study sequentially applied each modification to a custom guide rail base plate dataset. The backbone network was optimized based on the YOLOv8n model, referred to as Method i; the GhostConv module was introduced, referred to as Method ii; a small object detection layer was added, referred to as Method iii; and the CBAM was incorporated, referred to as Method iv. Ablation experiments were designed and conducted for comparative validation, with the results presented in Table 3 and Figure 12.

Based on the results of the ablation experiments, the sequential introduction of each module led to varying degrees of improvement in model performance. First, the MobileNetV3 module was introduced to optimize the backbone network, and the GhostConv module was applied to optimize the neck network. Although the precision and recall slightly decreased, with a minor reduction in mAP, the parameter count and GFLOPS were reduced, indicating that the optimization effectively decreased computational resource consumption. In comparison to the individual introduction of the modules, the simultaneous introduction of both modules led not only to a significant performance improvement but also to a notable reduction in network parameters. Subsequently, the addition of the small object detection layer led to a significant increase in precision, and this improvement significantly enhanced the detection capability for small-sized defect targets. Finally, the addition of the CBAM enabled the model to achieve optimal performance in terms of precision (P = 98.7%), recall (R = 98.9%), and mAP (99.4%), further optimizing the model’s ability to focus on key features during detection. Ultimately, compared to the original baseline model, the proposed improvements led to a 2.1% increase in mAP, a 35% reduction in the number of parameters, a 25.4% improvement in inference speed, and a 26.4% decrease in GFLOPs. These results robustly validate the effectiveness of the introduced enhancement modules.

The images in Figure 13 provide a clear visualization of the detection performance for tiny hole defects after the integration of each module. The detection image had a resolution of 1276 × 1276 pixels, with large hole defects measuring 26 × 26 pixels and small hole defects measuring 15 × 15 pixels. Prior to feature extraction, the model automatically resized the input image to 640 × 640 pixels, reducing the sizes of the large and small defects to 13 × 13 pixels and 7.5 × 7.5 pixels, respectively. Before the addition of the P2 detection layer, the model consistently failed to identify small-sized defects, which strongly corroborates the conclusion presented in Section 3.1: “tiny targets with areas smaller than 8 × 8 pixels lose their feature information in the P3 layer during downsampling.” However, after the incorporation of the P2 detection layer, and especially after the integration of the CBAM, the model not only successfully detected small-sized defects but also achieved a notable increase in the confidence levels of the prediction boxes. This experiment unequivocally demonstrates the critical role of the P2 detection layer and the CBAM in enhancing the model’s accuracy in detecting tiny hole defects.

### 4.5. Comparative Experiments of Different Detection Models

In this experiment, the proposed improved model was compared with lightweight networks from the YOLO series to further validate its superior performance in detecting defects in guide rail base plates. Table 3 presents a comparison of YOLOv5, YOLOv7, YOLOv8, YOLOv9, YOLOv10, YOLO11, and the proposed improved model across four metrics: precision (P), recall (R), mAP@0.5, and parameters.

From the comparison results in Table 4, it is evident that the accuracy, recall, and mAP values of each detection model vary. The precision of YOLOv5s is 93.6%, but it has a relatively large parameter count of 7,025,023, resulting in high computational resource consumption. Both YOLOv7t and YOLOv8n show improvements in their precision and mAP, with YOLOv7t achieving a precision of 97.9%, while YOLOv8n demonstrates a better balance. YOLOv9n and YOLOv10n, however, show slightly lower precision and recall but have smaller parameter counts. YOLO11n shows improvements in precision and recall but still has a relatively large parameter count. In contrast, our model (Ours) achieves optimal performance in terms of precision (98.7%), recall (98.9%), and mAP (99.4%), with a parameter count of 2,412,118, maintaining low computational resource consumption.

Table 5 presents a performance comparison between LGR-Net and other mainstream lightweight networks. Retinanet and SSD-EfficientNet achieved mAP@0.5 scores of 68.27% and 77.8%, respectively, with model sizes of 147.2 MB and 97.1 MB, and inference speeds of 71.4 FPS and 112.4 FPS. In comparison, LGR-Net, with a compact model size of just 5.4 MB, attained an impressive mAP@0.5 of 99.4% and an inference speed of 219.81 FPS, showcasing significantly enhanced performance. Overall, the proposed LGR-Net model demonstrates outstanding performance and computational efficiency, providing a superior solution for lightweight defect detection tasks.

### 4.6. Visualization of Experimental Results

Figure 14 demonstrates the model’s effectiveness in detecting hole defects, where LGR-Net identified tiny defects that YOLOv8 failed to detect, further validating the efficacy of the added P2 detection layer for small defects. Figure 15 and Figure 16 illustrate the model’s performance in detecting scratch and crack defects, respectively. Compared to YOLOv8, LGR-Net provided more accurate localization of these defects, highlighting the role of the CBAM in achieving precise defect localization. Figure 17 showcases the model’s ability to detect wear defects. Unlike other defects, wear defects exhibit less distinct features, making them challenging for YOLOv8 to identify. However, LGR-Net, leveraging the CBAM, accurately detected these defects. The visualization results clearly demonstrate the superior detection performance of the proposed LGR-Net network. Compared to the YOLOv8 algorithm, LGR-Net identified more defects with higher accuracy, provided more precise localization, and achieved significantly higher confidence levels in detection boxes. These visual results robustly validate the feasibility of the proposed LGR-Net model.

### 4.7. Performance Analysis in Practical Applications

As shown in Figure 18, on a low-computing-power device equipped with an Intel Core i3-6006U CPU (2.00 GHz, 4 GB RAM), LGR-Net achieved an inference speed of 124 FPS (including preprocessing and postprocessing), significantly surpassing YOLOv8n’s 86 FPS, representing a 44% improvement in inference speed. Additionally, LGR-Net’s computational complexity was 5.7 GFLOPs, which was 29.6% lower than YOLOv8n’s 8.1 GFLOPs and 3.01M parameters, demonstrating its greater deployment flexibility on resource-constrained devices. In real-world industrial scenarios, detection accuracy can be affected in environments with significant lighting variations. However, LGR-Net exhibits strong robustness against noise interference, even when Gaussian noise is added. These results indicate that LGR-Net not only operates efficiently on low-computing-power devices but also adapts well to complex industrial environments, showcasing its potential for deployment in actual production lines. This provides an efficient and cost-effective solution for industrial defect detection.

## 5. Conclusions

This paper proposes a lightweight rail base plate defect detection network (LGR-Net) based on YOLOv8 to address the issue of low accuracy in small target defect detection for rail base plates. The method used in this paper, which incorporates MobileNetV3 as the backbone network and optimizes the neck network using GhostConv convolutional modules, significantly reduces the parameter count of the baseline model, thereby reducing the computational overhead. After the addition of the small object detection layer and the CBAM, the network’s detection accuracy for small defects in rail base plates improved. The experimental results indicated that the final parameter count of LGR-Net was 2,412,118, which represents a 31.4% reduction compared to the original YOLOv8n. The final mAP@0.5 of the model reached 99.4%, indicating a relative improvement of 2.1%. LGR-Net maintains high detection accuracy while significantly reducing the computational cost, providing a feasible solution for the practical deployment of small defect detection in rail base plates.

## Figures and Tables

**Figure 1 sensors-25-01702-f001:**
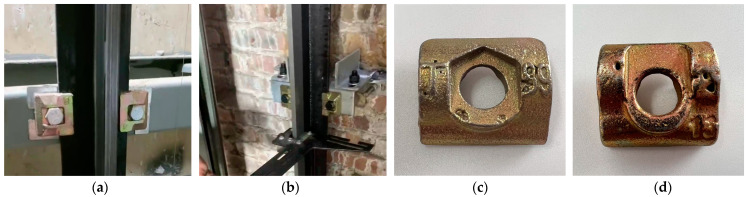
Schematic diagram of the guide rail clamp.

**Figure 2 sensors-25-01702-f002:**
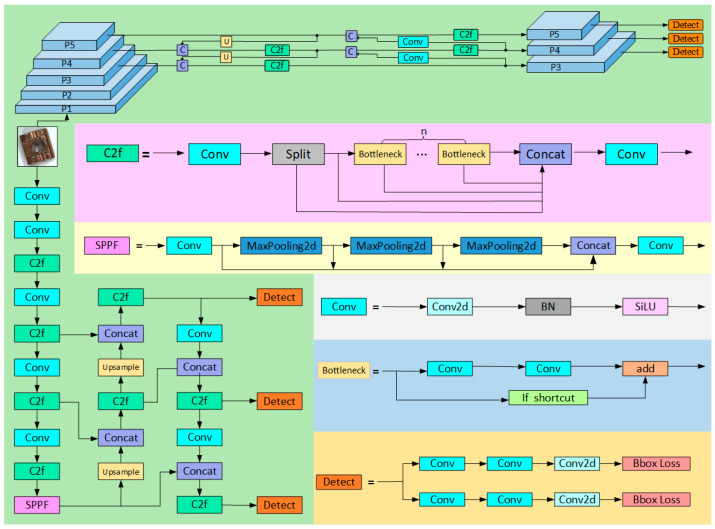
YOLOv8 network architecture diagram.

**Figure 3 sensors-25-01702-f003:**
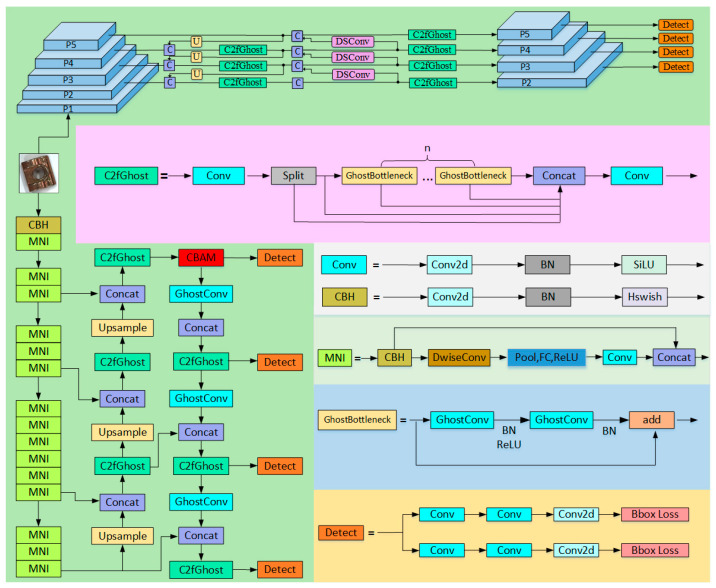
LGR-Net network structure.

**Figure 4 sensors-25-01702-f004:**
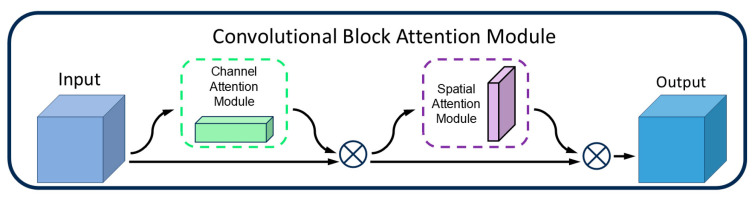
CBAM structure. Where ⊗ denotes element-wise multiplication.

**Figure 5 sensors-25-01702-f005:**
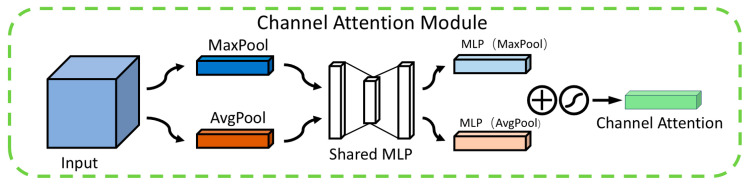
CAM structure. 
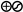
 represents element-wise addition and the application of the activation function sigmoid, respectively.

**Figure 6 sensors-25-01702-f006:**
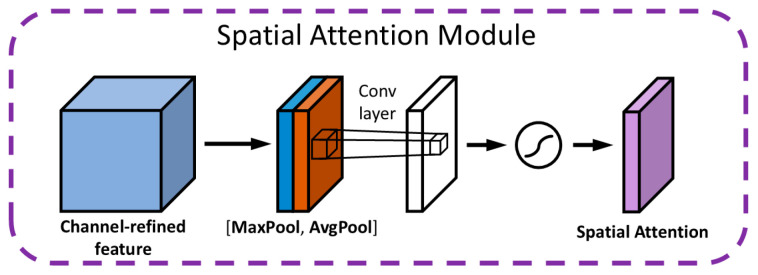
SAM structure. 
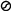
 represents the activation function Sigmoid.

**Figure 7 sensors-25-01702-f007:**
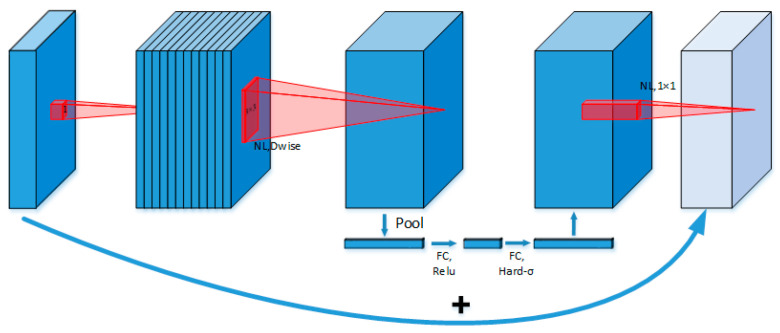
MobileNetV3 structure.

**Figure 8 sensors-25-01702-f008:**
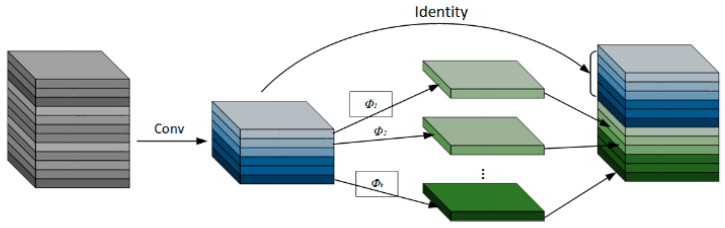
GhostConv structure.

**Figure 9 sensors-25-01702-f009:**
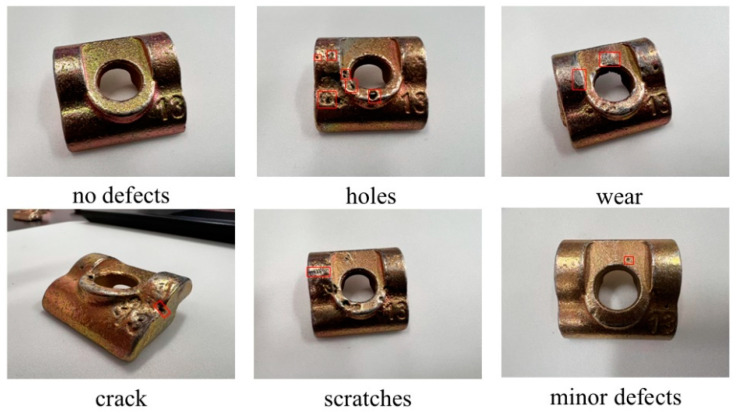
Defect types.

**Figure 10 sensors-25-01702-f010:**
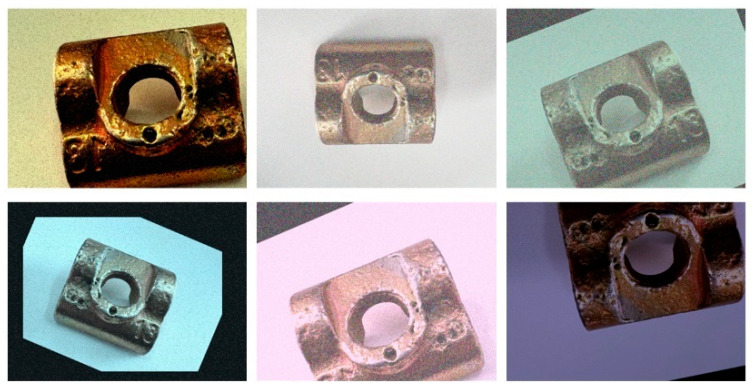
Images after data augmentation.

**Figure 11 sensors-25-01702-f011:**
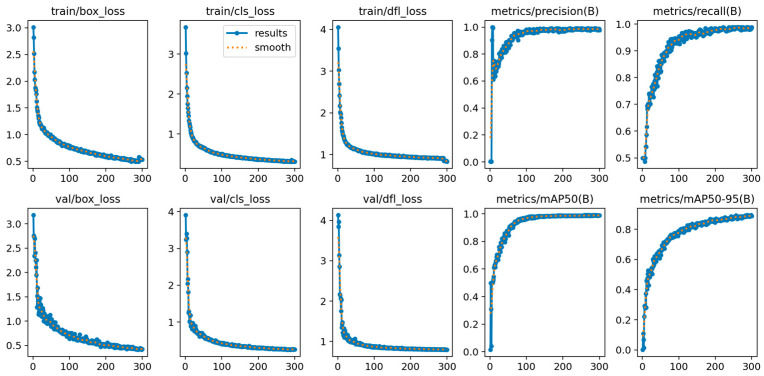
LGR-Net training results.

**Figure 12 sensors-25-01702-f012:**
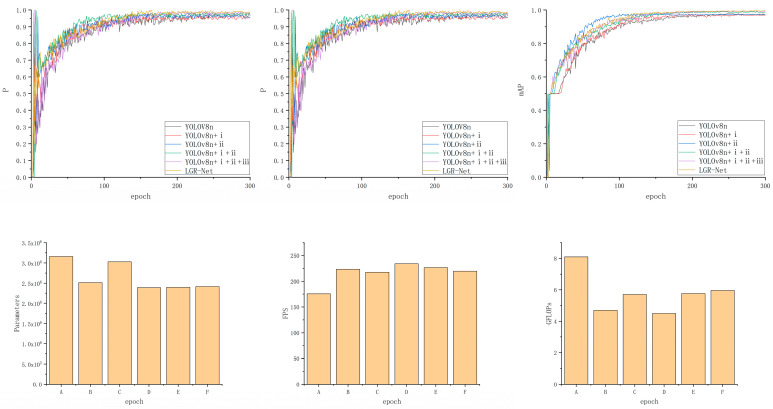
Visualization of ablation experiment data. ((A)—YOLOv8n, (B)—YOLOv8n + i, (C)—YOLOv8n + ii, (D)—YOLOv8n + i + ii, (E)—YOLOv8n + i +ii + iii, (F)—LGR-Net).

**Figure 13 sensors-25-01702-f013:**
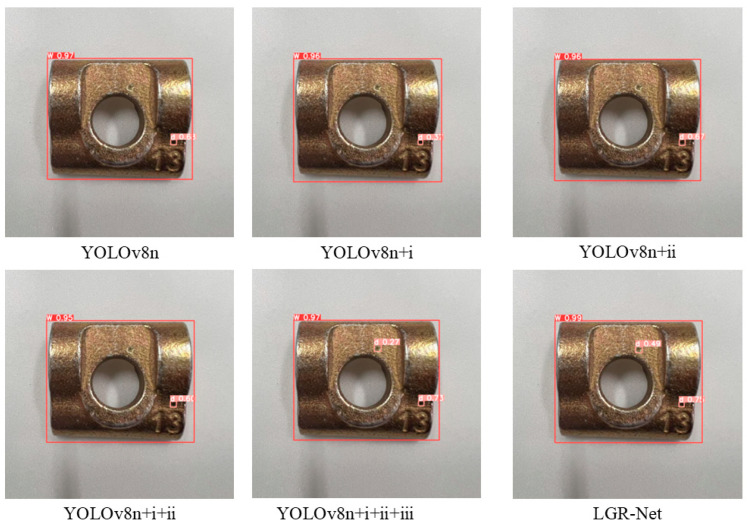
Visualization of detection results from ablation experiments.

**Figure 14 sensors-25-01702-f014:**
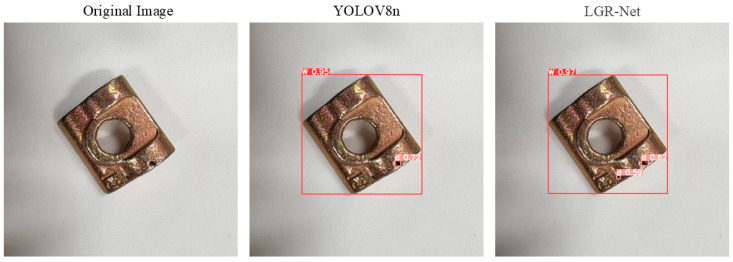
Detection results for small hole defects.

**Figure 15 sensors-25-01702-f015:**
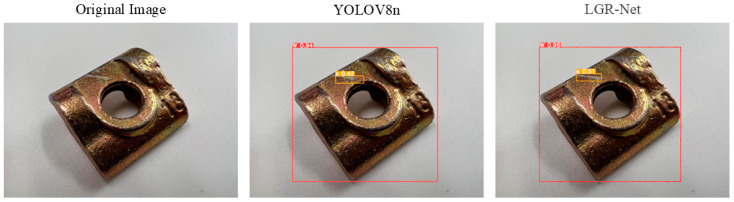
Detection results for scratch defects.

**Figure 16 sensors-25-01702-f016:**
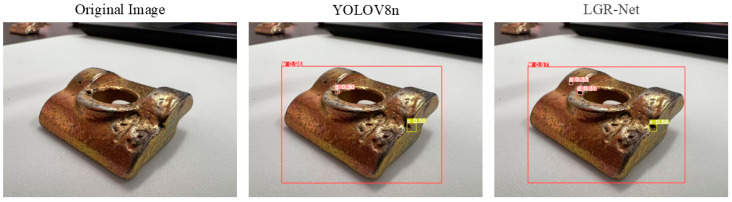
Detection results for crack defects.

**Figure 17 sensors-25-01702-f017:**
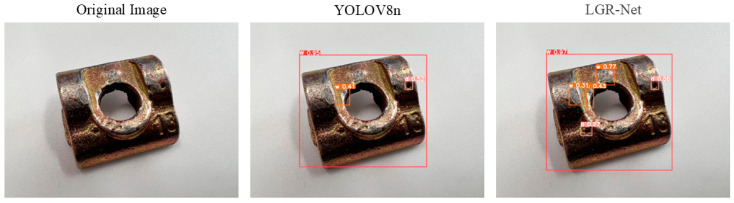
Detection results for wear defects.

**Figure 18 sensors-25-01702-f018:**
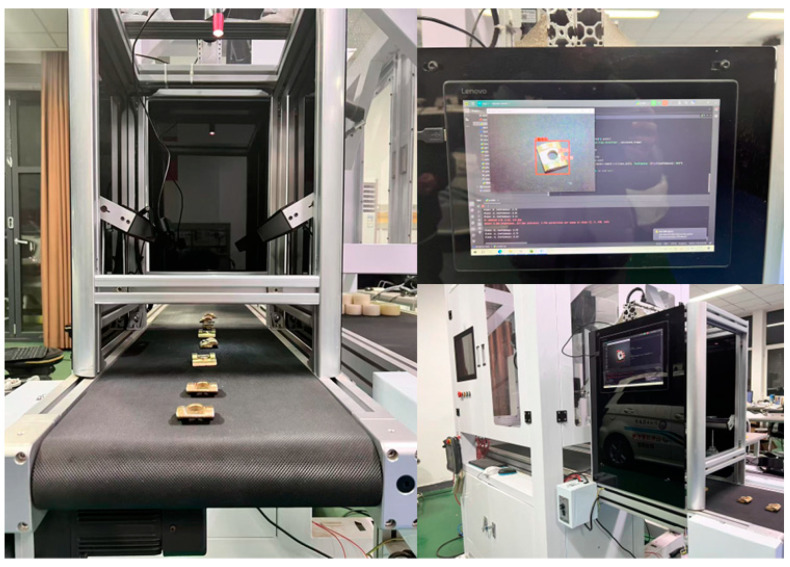
LGR-Net deployment demonstration.

**Table 1 sensors-25-01702-t001:** Experimental environment setup.

Parameter	Value
CPU	Intel^®^ Core™ i7-13700K, 16 cores, 24 threads, 5.40 GHz
GPU	NVIDIA GeForce RTX 3080 Laptop GPU
Operating System	Ubuntu 20.04
Torch Version	2.4.1 + cu121
Programming Language	Python 3.9.20

**Table 2 sensors-25-01702-t002:** Experimental parameter settings.

Parameter	Value
Iterations	300
Batch Size	16
Image Size	640 × 640
Initial Learning Rate	0.01
Momentum Factor	0.937
Weight Decay Coefficient	0.0005
Optimizer	SGD
Loss Function	CloU

**Table 3 sensors-25-01702-t003:** Ablation experiment results.

Model	P/%	R%	mAP@0.5/%	Parameters	FPS	GFLOPs
YOLOv8n	96.8	95.1	97.3	3,157,200	175.35	8.1
YOLOv8n + i	95.7	94.6	96.8	2,515,940	223.41	4.7
YOLOv8n + ii	96.1	94.8	97.1	3,027,551	217.75	5.7
YOLOv8n + i + ii	97.7	99.0	98.8	2,386,291	234.28	4.5
YOLOv8n + i + ii + iii	98.1	98.5	99.2	2,394,468	226.77	5.76
LGR-Net	98.7	98.9	99.4	2,412,118	219.81	5.96

**Table 4 sensors-25-01702-t004:** Comparative experiment results for different detection models.

Model	P/%	R%	mAP@0.5/%	Params (MB)	FPS	GFLOPs
YOLOv5s	93.6	95.5	96.8	7,025,023	157.31	16.5
YOLOv7t	97.9	94.0	97.2	6,017,694	161.73	10.1
YOLOv8n	96.8	95.1	97.3	3,157,200	175.35	8.1
YOLOv9n	96.4	94.5	97.1	2,659,676	183.54	7.7
YOLOv10n	93.7	94.7	97.4	2,707,820	217.79	8.2
YOLO11n	97.7	94.8	97.5	2,624,080	203.56	7.3
Ours	98.7	98.9	99.4	2,412,118	219.81	5.96

**Table 5 sensors-25-01702-t005:** Comparison with other mainstream lightweight networks.

Model	mAP@0.5/%	Params (MB)	FPS	GFLOPs
Retinanet	68.27	147.2	71.4	12.31
SSD-EfficientNet	77.8	97.1	112.4	11.74
Ours	99.4	5.4	219.81	5.96

## Data Availability

Data is contained within the article.

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
