# Peer review of "LGR-Net: A Lightweight Defect Detection Network Aimed at Elevator Guide Rail Pressure Plates"

_sensors, 2025, doi:10.3390/s25061702_

Round 1
Reviewer 1 Report
Comments and Suggestions for Authors
In this manuscript,a lightweight defect detection network(LGR-Net)based on the YOLOv8n algorithm is proposed for achieving high localization accuracy and computational efficiency when detecting small defects (e.g., cracks, scratches, wear) in guide rail pressure plates. The main contribution is as follows:
1) To reduce the excessive model parameters in the original algorithm, the baseline model's backbone network is enhanced by incorporating the lightweight MobileNetV3 and the neck network is optimized using the Ghost convolution module.
2) To improve localization accuracy for small defects, a high-resolution small-object detection layer (P2 layer) is added and the convolutional block attention module is integrated to construct a four-scale feature fusion network.
3) The experimental results based on a custom dataset show good defect detection performance in precision, recall, mAP and parameter count, achieving low computational complexity and high detection accuracy,providing an efficient and effective solution for defect detection in elevator guide rail pressure plates.
There have 2 suggestions as follows:
1) Give the typical pictures of cracks, scratches and wear.
2) Can give the detection results of cracks, scratches and wear respectively.
Author Response
1 Give the typical pictures of cracks, scratches and wear.
2 Can give the detection results of cracks, scratches and wear respectively.
Response:
Thank you for your careful review. We apologize for our oversight. Based on your comments, we have included the typical images of cracks, scratches, and wear in Section 4.1 and provided the detection results for cracks, scratches, and wear separately in Section 4.6. Please refer to Figures 9 and 14-17 for details
Reviewer 2 Report
Comments and Suggestions for Authors
(1) The paper presents an improved defect detection network (LGR-Net) based on YOLOv8 with modifications such as MobileNetV3, GhostConv, and CBAM. However, the specific novelty beyond integrating existing techniques is not clearly articulated. The authors should explicitly highlight how LGR-Net significantly differs from and advances beyond prior works.
(2) The study compares LGR-Net with other YOLO models, but it lacks comparisons with alternative lightweight defect detection networks outside the YOLO family (e.g., EfficientDet, RetinaNet, or transformer-based approaches). Including such comparisons would strengthen the evaluation.
(3) The paper emphasizes the efficiency of LGR-Net but does not discuss real-world implementation aspects such as inference speed on edge devices, hardware constraints, or deployment challenges in industrial settings. Addressing these factors would improve the paper’s practical relevance.
(4) The ablation experiments show performance improvements with individual modifications. However, a deeper analysis of how each component (e.g., CBAM, P2 layer) affects different defect types (small vs. large defects) would be valuable. A visualization of feature maps before and after incorporating these modules could further illustrate their impact.
Author Response
- The paper presents an improved defect detection network (LGR-Net) based on YOLOv8 with modifications such as MobileNetV3, GhostConv, and CBAM. However, the specific novelty beyond integrating existing techniques is not clearly articulated. The authors should explicitly highlight how LGR-Net significantly differs from and advances beyond prior works.
Response:
Thank you for your careful review. We apologize for our oversight. Based on your comments, we have explicitly highlighted how LGR-Net significantly differs from and advances beyond prior works in the final paragraph of Section 1 (Introduction).
- The study compares LGR-Net with other YOLO models, but it lacks comparisons with alternative lightweight defect detection networks outside the YOLO family (e.g., EfficientDet, RetinaNet, or transformer-based approaches). Including such comparisons would strengthen the evaluation.
Response:
Thank you for your careful review. We apologize for our oversight. Based on your comments, we have added comparisons with other lightweight defect detection networks (such as SSD-EfficientNet and RetinaNet) in Section 4.5. Please refer to Table 5 for details.
- The paper emphasizes the efficiency of LGR-Net but does not discuss real-world implementation aspects such as inference speed on edge devices, hardware constraints, or deployment challenges in industrial settings. Addressing these factors would improve the paper’s practical relevance.
Response:
Thank you for your careful review. We apologize for our oversight. Based on your comments, we have added a discussion in Section 4.7 regarding the deployment of LGR-Net on edge devices, as well as some practical challenges encountered in real-world applications.
- The ablation experiments show performance improvements with individual modifications. However, a deeper analysis of how each component (e.g., CBAM, P2 layer) affects different defect types (small vs. large defects) would be valuable. A visualization of feature maps before and after incorporating these modules could further illustrate their impact.
Response:
Thank you for your careful review. We apologize for our oversight. Based on your comments, we have provided detailed explanations in the text regarding the roles of each component (e.g., CBAM, P2 layer) in detecting both large and small defects. Please refer to Sections 3.1 and 3.2, as well as Figure 13, for further details.
Reviewer 3 Report
Comments and Suggestions for Authors
1.It is recommended to visually illustrate what a guide rail pressure plate is in the introduction using images.
2.The improvements mentioned in the introduction (e.g., MobileNetV3, GhostConv, CBAM, P2 layer) are too scattered. It is suggested to systematically explain how each module addresses specific problems and demonstrates its innovation. Additionally, provide a natural transition after summarizing existing issues to enhance logical coherence when introducing the research objectives.
3.The related work section covers a broad time range but rarely mentions significant advancements after 2022. It is recommended to include recent research on defect detection, lightweight networks, and attention mechanisms.
4.The article does not specify the loss function used during training. It is suggested to include this information to make the model design more complete.
5.The dataset section is overly simplistic. Consider using images to detail the categories and scenarios in the dataset, emphasizing its diversity and representativeness.
6.Comparative experiments only include YOLO series models. It is recommended to add comparisons with other mainstream lightweight networks, such as EfficientNet and ShuffleNet, to verify the model's generalizability.
7.The ablation studies provide only numerical results and lack visual analysis. It is suggested to include visualizations of detection results to demonstrate the impact of each module and further analyze cases of false positives and false negatives.
8. The experimental results in Section 4.6 are unclear and lack comparisons between original images and ground truths. It is recommended to include clearer images and further discuss the causes of detection failures.
9. As a lightweight network study, the article focuses solely on parameter count, lacking analysis of inference speed and FLOPs. It is suggested to include these metrics to comprehensively demonstrate the lightweight characteristics of the model.
Author Response
1. It is recommended to visually illustrate what a guide rail pressure plate is in the introduction using images.
Response:
Thank you for your careful review. We apologize for our oversight. Based on your comments, we have included visual illustrations of the guide rail pressure plate's appearance and function in the introduction section. Please refer to the first
paragraph of Section 1 and Figure 1 for details.
2. The improvements mentioned in the introduction (e.g., MobileNetV3, GhostConv,CBAM, P2 layer) are too scattered. It is suggested to systematically explain how each module addresses specific problems and demonstrates its innovation.
Additionally, provide a natural transition after summarizing existing issues to enhance logical coherence when introducing the research objectives.
Response:
Thank you for your careful review. We apologize for our oversight. Based on your comments, we have systematically explained how each module addresses specific problems at the end of the introduction section (Section 1).
3. The related work section covers a broad time range but rarely mentions significant advancements after 2022. It is recommended to include recent research on defect detection, lightweight networks, and attention mechanisms.
Response:
Thank you for your careful review. We apologize for our oversight. Based on your comments, we have supplemented recent works post-2022 in Sections 2.1 and 2.2.
4. The article does not specify the loss function used during training. It is suggested to include this information to make the model design more complete.
Response:
Thank you for your careful review. We apologize for our oversight. Based on your comments, we have specified the loss function used during training in Section 4.2. Please refer to Table 2 for details.
5. The dataset section is overly simplistic. Consider using images to detail the categories and scenarios in the dataset, emphasizing its diversity and representativeness.
Response:
Thank you for your careful review. We apologize for our oversight. Based on your comments, we have provided a detailed description of the dataset in Section 4.1, including images that illustrate the categories and scenarios, highlighting its
diversity and representativeness.
6. Comparative experiments only include YOLO series models. It is recommended to add comparisons with other mainstream lightweight networks, such as EfficientNet and ShuffleNet, to verify the model's generalizability.
Response:
Thank you for your careful review. We apologize for our oversight. Based on your comments, we have added comparisons with other lightweight defect detection networks (such as SSD-EfficientNet and RetinaNet) in Section 4.5. Please refer to
Table 5 for details.
7. The ablation studies provide only numerical results and lack visual analysis. It is suggested to include visualizations of detection results to demonstrate the impact of each module and further analyze cases of false positives and false negatives.
Response:
Thank you for your careful review. We apologize for our oversight. Based on your comments, we have added visual analysis of the numerical results in Section 4.4, including visualizations of detection results, and further discussed the reasons for
detection failures.
8. The experimental results in Section 4.6 are unclear and lack comparisons between original images and ground truths. It is recommended to include clearer images and further discuss the causes of detection failures.
Response:
Thank you for your careful review. We apologize for our oversight. Based on your comments, we have added clearer images in Section 4.6 and further analyzed the causes of detection failures.
9. As a lightweight network study, the article focuses solely on parameter count, lacking analysis of inference speed and FLOPs. It is suggested to include these metrics to comprehensively demonstrate the lightweight characteristics of the
model.
Response:
Thank you for your careful review. We apologize for our oversight. Based on your comments, we have included the inference speed (FPS) and GFLOPs in Sections 4.4 and 4.5.
Round 2
Reviewer 3 Report
Comments and Suggestions for Authors
I don't have any more questions